# The Impact of the COVID-19 Pandemic on Coronary Artery Bypass Grafting Surgery: A Single-Centre Retrospective Cohort Study

**DOI:** 10.3390/biomedicines13092264

**Published:** 2025-09-14

**Authors:** Paweł Nadziakiewicz, Marta Wajda-Pokrontka

**Affiliations:** Department of Anesthesiology and Intensive Therapy, Silesian Centre for Heart Diseases, Medical University of Silesia, 40-055 Katowice, Poland; marta.wajda93@gmail.com

**Keywords:** SARS-CoV-2 infection, coronary artery disease, cardiac surgical procedures, myocardial revascularisation, cardiothoracic surgery, cardiac surgery outcomes, pandemic impact on healthcare

## Abstract

**Background/Objectives:** The coronavirus disease 2019 (COVID-19) pandemic significantly impacted cardiac surgery, limiting patient access and altering care quality. This study evaluates changes in cardiovascular disease severity, types, and postoperative complications in patients qualifying for coronary artery bypass grafting (CABG) during the pandemic. **Methods:** We performed a retrospective analysis of 1499 CABG patients at our institution between March 2018 and February 2022. Patients were categorised into two groups: pre-pandemic (March 2018 to February 2020, N = 853) and pandemic (March 2020 to February 2022, N = 646). We analysed and detailed data across three major COVID-19 waves in Poland. **Results:** During the COVID-19 pandemic, 646 patients underwent CABG, a 24.3% decline from 853 pre-pandemic procedures. Urgent procedures increased from 37.6% to 44%, and life-saving procedures rose from 2.9% to 5.2% (*p* < 0.05). The use of cardiopulmonary bypass increased, along with longer procedure times (median of 279.7 min vs. 315 min; *p* < 0.001). The duration of mechanical ventilation increased during the pandemic period (median 12 h vs. 11 h; *p* < 0.05). No significant differences in postoperative complications were noted. Analysis during the three COVID-19 waves showed consistent baseline characteristics. In the second wave, life-saving CABG procedures reached 11.4%, with 17.5% of patients presenting acute coronary symptoms. **Conclusions:** The COVID-19 pandemic reduced CABG procedures, prioritising urgent cases. Short-term mortality odds rose, despite unchanged patient risk profiles. More multicentre research is needed to understand resource constraints on cardiac surgical outcomes and to establish guidelines for patient prioritisation in future pandemics.

## 1. Introduction

The coronavirus disease 2019 (COVID-19) pandemic, caused by the highly infectious severe acute respiratory syndrome coronavirus 2 (SARS-CoV-2), had a severe impact on healthcare systems worldwide due to the limited availability of treatments and vaccines [1]. The first COVID-19 case in Poland was detected on 4 March 2020 [2]. Since then, three major pandemic waves had been noted, forcing significant modifications to our national health system [3,4]. The pandemic prompted emergency restructuring of healthcare facilities to manage surges in critically ill patients, protect healthcare workers, and ensure access to crucial intensive care unit (ICU) resources while minimising SARS-CoV-2 transmission [2,5,6]. 

Focus during the pandemic shifted to primarily crisis management and critical care, limiting information about surgical services and patient outcomes. Cardiac surgery, in particular, experienced significant disruption due to its resource-intensive nature and the need for postoperative ICU admission [7,8]. A global survey of 60 cardiac surgery centres revealed a 50% to 75% reduction in cardiac surgeries during the pandemic, accompanied by a more than 50% decrease in dedicated cardiac operating rooms and ICU beds [9]. Cardiac surgical patients not only rely on critical ICU resources but also belong to the highest-risk category for COVID-19 complications, with a 30-day mortality rate estimated at 34% [10,11]. This was a key factor in delaying elective operations, particularly during the initial wave of the pandemic and before vaccines became available [12,13]. However, it remained a dilemma that existing heart disease also increased individual risks for hospitalisation or a severe course of the disease [14,15].

Currently, there is limited data on cardiac surgery patients during the pandemic in Poland compared with the pre-COVID-19 period [16,17]. No focused analysis has been conducted on coronary artery bypass grafting (CABG) patients in Poland during this time. This paper aimed to assess the impact of the COVID-19 pandemic on the provision of CABG surgery at our institution.

## 2. Materials and Methods

### 2.1. Study Population and Data Collection

This retrospective study comprised 1499 consecutive patients undergoing coronary artery bypass graft surgery at the Silesian Centre for Heart Diseases in Zabrze, Poland, between March 2018 and February 2022. Exclusions were made for lack of clinical data, composite procedures, and reoperations. The median age was 67 years (range 34 to 88 years). The patient population comprised 1125 (75.1%) men and 374 (24.9%) women.

The COVID-19 pandemic in Poland lasted from 20 March 2020 to 16 May 2022 [2,18]. This timeframe was used to categorise the studied population based on the dates of their surgical procedures. The pre-pandemic group consisted of patients who underwent surgery between March 2018 and February 2020 (N = 853, 32 women, 98 men), while the pandemic group was defined as those operated on from March 2020 to February 2022 (N = 646, 26 women, 120 men). Additionally, we specified and compared data from three major waves of the COVID-19 pandemic in Poland: the first, second, and third, which covered the periods from March 2020 to September 2020, October 2020 to February 2021, and March 2021 to February 2022, respectively [3,4].

We collected baseline demographics, comorbidities, operative urgency, New York Heart Association (NYHA) and Canadian Cardiovascular Society (CCS) classifications, as well as the European System for Cardiac Operative Risk Evaluation (EuroSCORE) II [19]. A critical preoperative state was defined by the presence of one or more of the following factors: intravenous unfractionated heparin therapy, use of inotropes, intravenous nitrates, or mechanical ventilation or mechanical circulatory support before surgery. Intraoperative data examined the application and duration of cardiopulmonary bypass (CPB); the number of procedures free from CPB, using off-pump coronary artery bypass (OPCAB) or minimally invasive coronary artery bypass (MIDCAB); as well as the number of performed bypass vessels. The definition of postoperative low cardiac output syndrome included one or more of the following: inotropic drugs, phosphodiesterase inhibitors, methylene blue, or mechanical circulatory support therapy. Examined outcomes included perioperative myocardial infarction; postoperative atrial fibrillation; postoperative renal replacement therapy implementation; duration of mechanical ventilation; the need to use mechanical circulatory support after surgery; neurological, septic, and pulmonary complications; reoperation; length of ICU and hospitalisation stay; and 30-day and in-hospital mortality. Data on long-term survival were obtained based on the Polish National Registry of Cardiac Surgical Operations database (KROK) and the Polish National Health Fund. We show all causes of mortality without differentiation of cardiac reasons.

### 2.2. Statistical Analysis

Results are reported as mean ± standard deviation or median (25th–75th percentile), depending on the variable distribution, for continuous variables. Categorical variables are presented as percentages (%). The distribution type was verified using the Smirnov–Kolmogorov test. Pre-pandemic and pandemic variables were equated using the Student’s *t*-test or the Mann–Whitney U test for continuous data and the chi-squared test for categorical ones. Variables from individual waves were compared using Pearson’s chi-squared test for categorical variables and the Kruskal–Wallis test for continuous variables. The odds ratios (ORs) and 95% confidence intervals (95% CIs) were calculated to determine the association between observation period (pandemic vs. pre-pandemic) and 30-day and in-hospital mortality. The survival analysis was conducted following the Kaplan–Meier methods, and the log-rank test was employed to compare the survival curves. A *p*-value of less than 0.05 was considered statistically significant. Statistical analysis was conducted using Statistica 13.3 (StatSoft, Inc., Tulsa, OK, USA).

## 3. Results

### 3.1. Pre-Pandemic vs. Pandemic Group

#### 3.1.1. Number of CABG Procedures and Basic Characteristics

The analysis involved 1499 patients, comprising 853 in the pre-pandemic group and 646 in the pandemic group, resulting in a 24.3% decline in CABG procedures. This amounts to 1.17 procedures per day before and 0.88 per day during the pandemic. Demographic characteristics, including sex and age, showed no statistically significant differences between the two groups [Table 1].

During the COVID-19 pandemic, the prevalence of patients with diabetes mellitus (DM) increased from 42.2% to 49.7%; *p* = 0.004. Similarly, differences were observed in arterial hypertension (HA) (90.7% vs. 85.1%, respectively; *p* = 0.001) and metabolic syndrome (16.7% vs. 12.8%, respectively; *p* = 0.032). Active and past nicotine abuse was more common in the pre-pandemic group (29.4% vs. 26.7%; *p* = 0.029 and 32.7% vs. 26.5%; *p* = 0.007, respectively). Comorbidities like chronic kidney, cerebrovascular, and pulmonary diseases were similar across both groups. However, during the pandemic, patients showed a lower incidence of CCS class IV (12.7% compared with 22%; *p* = 0.0001) and a reduced rate of preoperative critical states (17.6% versus 24.5%; *p* = 0.001). EUROSCORE II values remained similar between the groups.

#### 3.1.2. Surgical Data

Surgical data showed an increase in urgent procedures from 37.6% to 44% (*p* = 0.013) and life-saving procedures from 2.9% to 5.2% (*p* = 0.022) during the pandemic. The number of OPCAB procedures decreased from 49.7% to 44% (*p* = 0.027), and MIDCAB procedures dropped from 5.4% to 2.9% (*p* = 0.021) [Table 2].

The length of the procedure along with the duration of CPB was observed to be prolonged during the pandemic era, with a median of 279.7 min (range 100 to 635) compared with a median of 315 min (range 75 to 632), and a median of 82 min (range 67 to 97) versus a median of 98 min (range 80 to 123), respectively, both with a significance level of *p* = 0.0001. The number of bypass grafts performed and the incidence of triple vessel coronary artery disease (TVCAD) remained consistent throughout the COVID-19 pandemic. Nevertheless, the proportion of TVCAD within the study population varied based on the urgency of the procedures. During the pandemic, elective procedures for TVCAD were infrequently conducted, with a percentage of 45.9% compared with 36.8% (*p* = 0.025). In contrast, this proportion shifted for patients deemed eligible for urgent or life-saving procedures, as there was a notable increase of 18% in life-saving surgeries performed involving TVCAD during this period.

#### 3.1.3. Outcome Data and Survival Analysis

The postoperative period showed no significant differences in renal, pulmonary, septic, or neurological complications. The rates of postoperative low cardiac output syndrome and the need for mechanical support after surgery were similar in both groups [Table 3].

The duration of mechanical ventilation during the pandemic increased to a median of 12 h (range 9.6 to 15), compared with 11 h (range 9.3 to 13.6; *p* = 0.003). None of the analysed patients in either era required non-invasive positive pressure ventilation post-extubation or tracheostomy postoperatively. ICU stays were longer during the pandemic, averaging 1.9 ± 3.3 days versus 1.7 ± 2.5 days (*p* = 0.029).

During the COVID-19 pandemic, the odds of mortality within 30 days after surgery were 2.35 (OR, 2.35; 95% CI, 0.96–3.67; *p* = 0.026) times higher, and the odds of in-hospital mortality were 1.88 (OR, 1.88; 95% CI, 1.15–4.82; *p* = 0.089) times higher, compared with patients who underwent surgery before the pandemic. The Kaplan–Meier curve estimated survival after CABG showed similar survival rates regardless of whether the procedure occurred before or during the pandemic era (log-rank test; *p* = 0.443) (Figure 1).

### 3.2. The Three Waves of the COVID-19 Pandemic

#### 3.2.1. Number of CABG Procedures and Basic Characteristics

The number of procedures performed varied during the COVID-19 pandemic, with 221 procedures in the first wave, 114 in the second wave, and 311 during the third wave. These figures represented an average of 1.04, 0.76, and 1.17 procedures per day in the first, second, and third waves, respectively, during each subsequent pandemic wave. Preoperative characteristics, including pulmonary, renal, and cerebrovascular comorbidities, as well as the prevalence of DM and HA, were similar across groups. However, hyperlipidaemia prevalence was significantly higher in the third wave (80.7%) compared with the first (72.4%) and second waves (71.9%; *p* = 0.041) [Table 4].

Patients in the third wave had a higher rate of NYHA class I classification (16.7%) compared with the first (8.1%) and second waves (7%), with a statistically significant difference (*p* = 0.002). In contrast, the second wave had more CCS class IV patients (17.5%) than the first (10.4%) and third waves (12.5%), but this did not reach statistical significance (*p* = 0.177).

#### 3.2.2. Surgical Data

Surgical data showed no significant difference in critical preoperative states (*p* = 0.179). However, second-wave patients underwent more life-saving procedures (11.4%) compared with the first (2.7%) and third waves (4.2%), which was significant (*p* = 0.003). The percentage of OPCAB procedures decreased over the waves: 52.5% in the first, 36.8% in the second, and 38.6% in the third (*p* = 0.002) [Table 5].

The MIDCAB procedures were infrequent across all waves (*p* = 0.499). Procedure duration increased from 300 min in the first wave to 330 min in the third wave (*p* = 0.0001), while CPB time remained stable (*p* = 0.469).

#### 3.2.3. Outcome Data and Survival Analysis

Early postoperative complications showed a rise in septic complications (3.5%) and neurological complications (6.1%) during the second wave of the COVID-19 pandemic, though these differences were not statistically significant [Table 6].

Conversely, there was a rise in pulmonary complications, with rates of 1.8% during the first wave, 7% during the second wave, and 5.1% during the last wave; *p* = 0.052. The duration of mechanical ventilation was similar across all waves (median 11.5, 11.9, and 11.8 h; *p* = 0.535). The median length of ICU stay and overall hospitalization remained unchanged, although the second wave had a longer average hospitalization (11.1 ± 19.2 days vs. 8.5 ± 5.8 days in the first wave and 7.9 ± 5.7 days in the third wave; *p* = 0.046) and longer ICU stays (3.3 ± 3.5 days; *p* = 0.020). No significant variation was observed in in-hospital or 30-day mortality across the waves.

## 4. Discussion

This study examines the impact of the COVID-19 pandemic on coronary artery bypass grafting procedures at a single centre in Poland. Over the two years of the pandemic, there was a 24.3% decrease in CABG procedures compared with the same time in previous years (March 2018–February 2020). The pandemic era was not a homogeneous period, and surgical activity changed simultaneously with the consecutive waves of the pandemic. The most significant decline in procedures occurred during the second wave, averaging 0.76 procedures per day. Interestingly, the first wave saw only a minor reduction to 1.07 procedures per day. This initial decline coincided with the comprehensive lockdown imposed by the Polish government [2]. Since the beginning of the third wave of the pandemic, the number of performed procedures has returned to the pre-pandemic baseline, reaching 1.17 CABG procedures daily. The observed reduction in CABG operating capacity was consistent with international experience, ranging from 2.3% to 60% [20,21,22,23,24]. The decline in the overall number of coronary artery bypass grafting procedures was directly linked to a reduction in hospitalisations due to acute coronary syndrome (ACS). A study by Mafham et al. [25] demonstrated a 40% (95% CI 37–43) decrease in hospital admissions for all types of acute coronary syndromes in England. Additionally, the reduced proportion of patients undergoing CABG in the studied population ranged between 75% and 80% (95% CI 33–91), depending on the specific category of ACS. Similar reports covered the Polish population with coronary artery disease during the COVID-19 pandemic. Bychowski et al. [26] observed a decrease in monthly admissions for STEMI cases, with averages dropping from 13.79 in the pre-pandemic group to 12.42 during the pandemic. A multi-institutional study conducted by Jankowska-Sanetra and colleagues [27] reported a 6.7% reduction in hospitalisations for ACS throughout the COVID-19 pandemic. Additionally, their findings indicated a decline in referrals for CABG from 4.9% to 3.7%.

The COVID-19 pandemic led to substantial changes in medical care delivery, with a shift from treating cardiac surgical patients to focusing on curing COVID-19 cases [11,28,29]. At the beginning of the COVID-19 pandemic outbreak, national cardiothoracic societies, including the Polish Society of Cardiothoracic Surgeons, recommended the cancellation of elective procedures [30,31]. The postponement of elective surgery in high-risk patients was primarily due to their increased risk of contracting COVID-19 and the higher likelihood of mortality and poor outcomes [32,33]. In our cohort study, the proportion of elective procedures decreased from 47% to 40.4% during the COVID-19 pandemic (*p* = 0.0107). During the same period, there was a significant increase in the number of life-saving procedures (5.2%, *p* = 0.022) and urgent procedures performed within 7 days (44%, *p* = 0.013). The most notable decline in elective procedures occurred during the second wave of the pandemic, with a 26.3% decrease (*p* = 0.0003). Moreover, the percentage of life-saving procedures increased rapidly to 11.4% (*p* = 0.003). Surprisingly, those findings were not associated with an overall severe risk profile of patients undergoing CABG during the whole pandemic period. The number of patients admitted in a critical preoperative state decreased during the COVID-19 era, from 24.5% to 17.6% (*p* = 0.001). The percentage of patients classified as CCS IV also decreased almost twofold (22% vs. 12.7%, *p* = 0.0001).

Several factors can be proposed to explain this observation. The analysis of preoperative data suggested that patients undergoing CABG procedures during the pandemic had a higher prevalence of diabetes (49.7% vs. 42.2%, *p* = 0.005) and metabolic syndrome (16.7% vs. 12.8%, *p* = 0.032). Moreover, compared with previous years, TVCAD rates in the pandemic group were higher, with an increasing urgency of procedures, particularly for life-saving procedures (58.8% vs. 40%, *p* = 0.244). These results suggest that patients undergoing CABG during the pandemic required urgent revascularisation due to specific patterns of coronary artery disease (CAD) [34]. The 2018 European Guidelines on Myocardial Revascularisation [35] included a recommendation for CABG over percutaneous coronary intervention (PCI) for CAD patients with diabetes or multivessel disease with the SYNTAX (Synergy between PCI with Taxus and Cardiac Surgery) score > 23 [36]. This notion was based on the FREEDOM (Future Revascularisation Evaluation in Patients with Diabetes Mellitus: Optimal Management of Multivessel Disease) trial [37]. The study revealed that patients who underwent CABG experienced significantly lower rates of the combined primary outcome, which includes death from any cause, myocardial infarction, or stroke, compared with those who had PCI with a drug-eluting stent. This finding was consistent across all levels of angiographic complexity, as assessed by the SYNTAX score, as well as concerning ejection fraction and renal function. However, it is essential to acknowledge that delays in the safe resumption of cardiac surgery services due to the COVID-19 pandemic likely influenced decision making regarding revascularisation strategies [38]. As a result, there was a shift towards PCI, which is associated with shorter hospital stays and the possibility of same-day discharges [39]. This approach aimed to prevent further increases in waiting lists for CABG. Kite et al. [40] demonstrated that PCI performed using contemporary techniques yielded comparable short-term outcomes in patients who would otherwise be candidates for CABG during the pandemic era.

Previous reports [23,41] indicated that patients undergoing coronary artery bypass grafting during the pandemic had significantly higher rates of morbidity, experienced more postoperative complications, and had a poorer prognosis. Despite a greater proportion of specific comorbidities, such as diabetes mellitus, hypertension, and metabolic syndrome as well as a higher rate of emergency surgeries and prolonged CPB durations, the outcomes after CABG surgery during the pandemic were comparable to those before the pandemic, which is unexpected at first sight. There was no statistically significant difference in the rates of postoperative low cardiac output or complications related to sepsis or pulmonary or neurological events. The only statistically significant differences observed in postoperative outcomes during the pandemic period pertained to the duration of mechanical ventilation, the length of stay in the ICU, and the total duration of hospitalisation. These findings were consistent with those reported by Wang et al. [41]. It is important to note that the expected mortality rate, as calculated by the EUROSCORE II score, was similar between the pre-pandemic and pandemic groups (2.2 [1.2; 4.6] vs. 1.99 [1.2; 4], *p* = 0.178). However, in-hospital mortality nearly doubled during the pandemic (1.8% vs. 3.3%, *p* = 0.089). Additionally, data analysis showed that patients who underwent coronary artery bypass grafting during the pandemic had higher odds of 30-day mortality (OR 2.35; 95% CI, 1.15–4.82; *p* = 0.017) compared with those before the pandemic. This evidence suggests that the 2.35-fold increase in the likelihood of 30-day death among individuals undergoing CABG during the pandemic is directly linked to the adverse effects of the COVID-19 era. Similar conclusions were drawn by Mejia and colleagues [23], who found a 2.8-fold increase in the mortality risk (95% CI, 1.0–7.6; *p* = 0.041) for CABG procedures performed in 2020 compared with those in 2019. In contrast, a study by Parcha et al. [21] reported no differences in the odds of mortality within 30 days (OR 0.96; 95% CI, 0.69–1.33) when comparing procedures from 2019 to those from 2020. It is essential to highlight that, within the analysed population, the long-term survival rate at the 4-year follow-up, as illustrated in the Kaplan–Meier graph, yielded comparable results across both groups (log-rank test, *p* = 0.433).

Several limitations in our study necessitate careful consideration. It is a retrospective, single-centre analysis, which makes it susceptible to bias and residual confounding. These attributes of design naturally limit the external validity and broad applicability of our findings. Variations in hospital capacity, regional pandemic severity, and local clinical protocols may have influenced surgical practices and outcomes elsewhere. Therefore, future multicentre or national data-based studies are crucial to validate our results and deepen the understanding of the pandemic’s impact on cardiac surgery services. The main limitation is the lack of precise data on individual patients’ COVID-19 infection statuses; therefore, the potential impact of perioperative COVID-19 infection on outcomes could not be evaluated. Moreover, the long-term outcome was not recorded in this study. Adverse effects of a more critical preoperative condition may only become apparent as long-term consequences for cardiovascular health. Additionally, our study may underestimate the impact of the COVID-19 pandemic on outcomes after cardiac surgery. Patients may have been unable to access life-saving cardiac surgical care due to pandemic-related barriers, and as a result, they are not included in our study. Furthermore, some patients who would typically have undergone cardiac surgery may have instead received percutaneous therapy due to limited inpatient capacity during the pandemic.

## 5. Conclusions

In conclusion, the COVID-19 pandemic had a negative impact on the volume of patients undergoing CABG procedures. The pandemic era was accompanied by a prioritisation of emergency procedures at the expense of elective surgeries. The pandemic era increased the odds of short-term mortality without worsening patients’ risk profiles, as indicated by the EUROSCORE II. To gain a comprehensive understanding of how limited healthcare resources affected cardiac surgery, interdisciplinary and multicentre research is necessary to provide evidence-based prioritisation of patients in future pandemics, thereby minimising the negative impact on all patients.

## Figures and Tables

**Figure 1 biomedicines-13-02264-f001:**
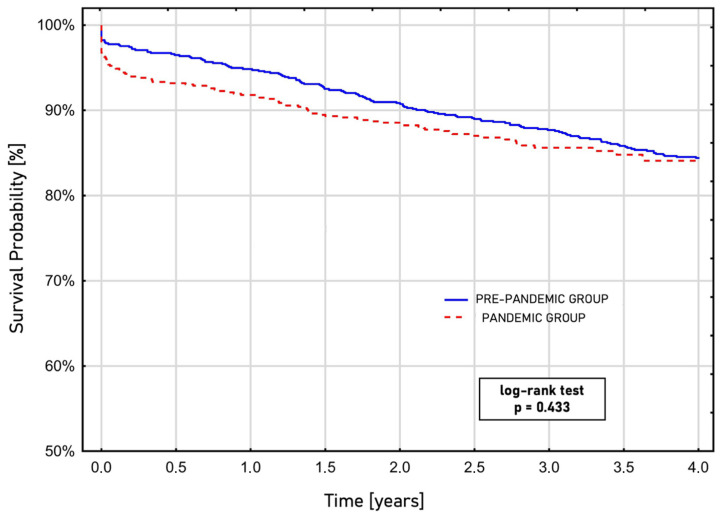
Long-term survival after coronary artery bypass grafting based on the performance time before or during the COVID-19 pandemic, estimated by the Kaplan–Meier curve.

**Table 1 biomedicines-13-02264-t001:** Preoperative clinical data of the study cohort.

	Pre-Pandemic N = 853	Pandemic N = 646	*p*-Value
Sex, men, N (%)	626 (73.4)	499 (77.2)	0.088
Age, years	67.3 [61.3; 72.4]	67.1 [61.7; 72.6]	0.670
Body mass index, kg/m^2^	28.7 [25.8; 31.5]	28.4 [25.4; 31.7]	0.789
Diabetes mellitus, N%	360 (42.2)	321 (49.7)	0.004
Hyperlipidaemia, N (%)	658 (77.1)	493 (76.3)	0.755
Arterial hypertension, N (%)	726 (85.1)	586 (90.7)	0.001
Active nicotine abuse, N (%)	241 (29.4)	157 (26.7)	0.029
Nicotine abuse in past, N (%)	279 (32.7)	171 (26.5)	0.007
Metabolic syndrome, N (%)	109 (12.8)	108 (16.7)	0.032
Chronic kidney disease, N (%)	110 (12.9)	96 (14.9)	0.308
Estimated glomerular filtration rate, mL/min/1.73 m^2^	77.2 [61.4; 91.5]	75.2 [58.3; 90.3]	0.134
Pulmonary comorbidities, N (%)	65 (7.6)	54 (8.4)	0.596
Cerebrovascular comorbidities, N (%)	102 (11.9)	63 (9.8)	0.177
CCS I, N (%)	36 (4.2)	36 (5.6)	0.176
CCS II, N (%)	358 (41.9)	284 (43.9)	0.188
CCS III, N (%)	224 (26.3)	169 (26.2)	0.751
CCS IV, N (%)	188 (22)	82 (12.7)	0.0001
NYHA I, N (%)	142 (16.6)	186 (28.8)	0.0001
NYHA II, N (%)	532 (62.4)	351 (54.3)	0.013
NYHA III, N (%)	163 (19.1)	98 (15.2)	0.085
NYHA IV, N (%)	16 (1.9)	11 (1.7)	0.867
Critical preoperative state, N (%)	209 (24.5)	114 (17.6)	0.001
EUROSCORE II, %	2.2 [1.2; 4.6]	1.99 [1.2; 4.0]	0.178
Ejection fraction, %	50 [43; 55]	50 [42; 55]	0.191

Values are expressed as mean ± SD, median [25th, 75th percentile]. Abbreviations: CCS, the Classifications of the Canadian Cardiovascular Society; NYHA, the Classifications of the New York Heart Association.

**Table 2 biomedicines-13-02264-t002:** Surgical data.

	Pre-Pandemic N = 853	Pandemic N = 646	*p*-Value
Elective procedure, N (%)	401 (47)	261 (40.4)	0.012
Urgency procedure to 30 days, N (%)	106 (12.5)	67 (10.4)	0.218
Urgency procedure to 7 days, N (%)	321 (37.6)	282 (44)	0.013
Life-saving procedure, N (%)	25 (2.9)	34 (5.2)	0.022
Use of cardiopulmonary bypass, N (%)	383 (44.9)	470 (53.1)	0.002
Type of procedure, MIDCAB, N (%)	46 (5.4)	19 (2.9)	0.021
Type of procedure, OPCAB, N (%)	424 (49.7)	284 (44)	0.027
Duration of procedure, min	279.7 [100; 635]	315 [75; 632]	0.0001
Duration of cardiopulmonary bypass, min	82 [67; 97]	98 [80; 123]	0.0001
Number of bypass vessels, N (%)	2 [2; 3]	2 [2; 3]	0.303
TVCAD, N (%)	411 (48.2)	298 (46.1)	0.462
Elective procedure with TVCAD, N (%)	184 (45.9)	96 (36.8)	0.025
Urgency procedure to 30 days with TVCAD, N (%)	49 (46.2)	28 (49.8)	0.678
Urgency procedure to 7 days with TVCAD, N (%)	168 (52.3)	154 (54.6)	0.634
Life-saving procedure with TVCAD, N (%)	10 (40)	20 (58.8)	0.244

Values are expressed as mean ± SD, median [25th, 75th percentile]. Abbreviations: MIDCAB, minimally invasive direct coronary artery bypass; OPCAB, off-pump coronary artery bypass; TVCAD, triple vessel coronary artery disease.

**Table 3 biomedicines-13-02264-t003:** Outcome data of the cohort study.

	Pre-Pandemic N = 853	Pandemic N = 646	*p*-Value
Perioperative myocardial infarction, N (%)	6 (0.7)	4 (0.6)	0.843
Postoperative low cardiac output syndrome, N (%)	238 (27.9)	194 (30)	0.367
Postoperative mechanical support, N (%)	16 (1.9)	12 (1.9)	0.979
Duration of mechanical ventilation, hours	11 [9.3; 13.6]	12 [9.6; 15]	0.003
Reintubation, N (%)	10 (1.2)	14 (2.2)	0.129
Postoperative atrial fibrillation, N (%)	38 (4.5)	31 (4.8)	0.753
Renal replacement therapy, N (%)	6 (2.7)	0	0.148
Reoperation, N (%)	14 (6.3)	6 (5.3)	0.856
Pulmonary complication, N (%)	29 (3.4)	28 (4.3)	0.349
Septic complication, N (%)	18 (2.1)	16 (2.5)	0.637
Neurologic complication, N (%)	30 (3.5)	26 (4)	0.608
Intensive care unit stay, days	1.7 ± 2.5	1.9 ± 3.3	0.029
Duration of hospitalization, N (%)	8.5 ± 5.6	8.7 ± 9.6	0.249
In-hospital mortality, N (%)	15 (1.8)	21 (3.3)	0.089
30-day mortality, N (%)	12 (1.4)	21 (3.3)	0.026

Values are expressed as mean ± SD, median [25th, 75th percentile].

**Table 4 biomedicines-13-02264-t004:** Preoperative clinical data in the three waves of the COVID-19 pandemic.

	1st Wave N = 221	2nd Wave N = 114	3rd Wave N = 311	*p*-Value
Sex, men, N (%)	162 (73.3)	92 (80.7)	245 (78.8)	0.208
Age, years	67 [61.9; 72.7]	66.3 [61.7; 71.8]	67.3 [61.5; 72.8]	0.624
Body mass index, kg/m^2^	28.4 [25.1; 31.8]	28.7 [25.4; 32.8]	28.4 [25.5; 31.2]	0.789
Diabetes mellitus, N%	110 (49.8)	52 (45.6)	159 (51.1)	0.602
Hyperlipidaemia, N (%)	160 (72.4)	82 (71.9)	251 (80.7)	0.041
Arterial hypertension, N (%)	203 (91.9)	98 (85.9)	288 (92.6)	0.092
Active nicotine abuse, N (%)	45 (20.4)	30 (26.3)	82 (26.4)	0.242
Nicotine abuse in past, N (%)	110 (49.8)	50 (43.9)	168 (54)	0.167
Metabolic syndrome, N (%)	36 (16.3)	14 (12.3)	58 (18.6)	0.294
Chronic kidney disease, N (%)	26 (11.8)	14 (12.3)	45 (14.5)	0.631
Estimated glomerular filtration rate, mL/min/1.73 m^2^	74.9 [60.1; 86.2]	81.2 [61.6; 98.2]	73.5 [55.6; 90.4]	0.062
Pulmonary comorbidities, N (%)	21 (9.5)	11 (9.6)	22 (7.1)	0.523
Cerebrovascular comorbidities, N (%)	25 (11.3)	11 (9.6)	27 (8.7)	0.601
CCS I, N (%)	11 (4.9)	5 (4.4)	18 (5.8)	0.825
CCS II, N (%)	103 (46.6)	46 (40.4)	135 (43.4)	0.469
CCS III, N (%)	61 (27.6)	32 (28.1)	76 (24.4)	0.803
CCS IV, N (%)	23 (10.4)	20 (17.5)	39 (12.5)	0.177
NYHA I, N (%)	18 (8.1)	8 (7)	52 (16.7)	0.002
NYHA II, N (%)	144 (65.2)	61 (53.5)	146 (46.9)	0.001
NYHA III, N (%)	30 (13.6)	20 (17.5)	48 (15.4)	0.359
NYHA IV, N (%)	2 (0.9)	1 (0.9)	7 (2.3)	0.378
Critical preoperative state, N (%)	44 (19.9)	24 (21.1)	46 (14.8)	0.179
EUROSCORE II, %	2.3 [1.3; 4.1]	1.9 [1.2; 4.3]	1.9 [1.2; 4]	0.212
Ejection fraction, %	50 [42; 55]	50 [42; 55]	50 [40; 55]	0.863

Values are expressed as mean ± SD, median [25th, 75th percentile]. Abbreviations: CCS, the Classifications of the Canadian Cardiovascular Society; NYHA, the Classifications of the New York Heart Association.

**Table 5 biomedicines-13-02264-t005:** Surgical data for the three waves of the COVID-19 pandemic.

	1st Wave N = 221	2nd Wave N = 114	3rd Wave N = 311	*p*-Value
Elective procedure, N (%)	84 (38.1)	30 (26.3)	148 (47.6)	0.0003
Urgency procedure to 30 days, N (%)	36 (16.2)	10 (8.8)	42 (13.5)	0.164
Urgency procedure to 7 days, N (%)	95 (42.3)	61 (53.5)	128 (41.2)	0.071
Life-saving procedure, N (%)	6 (2.7)	13 (11.4)	13 (4.2)	0.003
Use of cardiopulmonary bypass, N (%)	97 (43.9)	67 (58.8)	184 (59.2)	0.001
Type of procedure, MIDCAB, N (%)	7 (3.2)	5 (4.4)	7 (2.3)	0.499
Type of procedure, OPCAB, N (%)	116 (52.5)	42 (36.8)	120 (38.6)	0.002
Duration of procedure, min	300 [255; 345]	305 [260; 360]	330 [285; 380]	0.0001
Duration of cardiopulmonary bypass, min	92.5 [80; 121]	98 [78; 128]	102 [80; 125]	0.469
Number of bypass vessels, N (%)	2 [2; 3]	2 [2; 3]	2 [2; 3]	0.827
TVCAD, N (%)	97 (43.9)	54 (47.4)	147 (47.3)	0.954
Elective procedure with TVCAD, N (%)	30 (13.6)	9 (7.9)	57 (18.3)	0.107
Urgency procedure to 30 days with TVCAD, N (%)	15 (6.8)	3 (2.6)	10 (3.2)	0.293
Urgency procedure to 7 days with TVCAD, N (%)	50 (22.6)	35 (30.7)	69 (22.2)	0.462
Life-saving procedure with TVCAD, N (%)	2 (0.9)	7 (6.1)	11 (3.5)	0.123

Values are expressed as mean ± SD, median [25th, 75th percentile]. Abbreviations: MIDCAB, minimally invasive direct coronary artery bypass; OPCAB, off-pump coronary artery bypass; TVCAD, triple vessel coronary artery disease.

**Table 6 biomedicines-13-02264-t006:** Outcome data for the three waves of the COVID-19 pandemic.

	1st Wave N = 221	2nd Wave N = 114	3rd Wave N = 311	*p*-Value
Perioperative myocardial infarction, N (%)	1 (0.47)	0	4 (1.3)	0.690
Postoperative low cardiac output syndrome, N (%)	63 (28.5)	33 (28.9)	99 (31.8)	0.677
Postoperative mechanical support, N (%)	4 (1.8)	3 (2.6)	5 (1.6)	0.975
Duration of mechanical ventilation, hours	11.5 [9.3; 15]	11.9 [9.5; 15.5]	11.8 [9.8; 14.9]	0.535
Reintubation, N (%)	4 (1.8)	2 (1.8)	8 (2.6)	0.792
Postoperative atrial fibrillation, N (%)	14 (6.3)	4 (3.5)	13 (4.2)	0.769
Renal replacement therapy, N (%)	6 (2.7)	0	4 (1.3)	0.148
Reoperation, N (%)	14 (6.3)	6 (5.3)	21 (6.8)	0.856
Pulmonary complication, N (%)	4 (1.8)	8 (7)	16 (5.1)	0.053
Septic complication, N (%)	4 (1.8)	4 (3.5)	8 (2.6)	0.631
Neurologic complication, N (%)	4 (1.8)	7 (6.1)	15 (4.8)	0.098
Intensive care unit stay, days	2.6 ± 1.9	3.3 ± 3.5	2.7 ± 1.3	0.020
Duration of hospitalization, N (%)	8.5 ± 5.8	11.1 ± 19.2	7.9 ± 5.7	0.046
In-hospital mortality, N (%)	5 (2.3)	5 (4.4)	11 (3.5)	0.539
30-day mortality, N (%)	5 (2.3)	5 (4.4)	11 (3.5)	0.539

Values are expressed as mean ± SD, median [25th, 75th percentile].

## Data Availability

The raw data supporting the conclusions of this article will be made available by the authors upon request and institutional approval.

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
