# Peer review of "The Impact of the COVID-19 Pandemic on Coronary Artery Bypass Grafting Surgery: A Single-Centre Retrospective Cohort Study"

_biomedicines, 2025, doi:10.3390/biomedicines13092264_

Round 1

Reviewer 1 Report

Comments and Suggestions for Authors

The authors reported a single-center retrospective cohort study from Poland, examining the distribution and outcomes of patients undergoing CABG before and during the COVID-19 pandemic.

I have several comments and questions.

  1. Page 4, line 127: The statement “patients during the pandemic had a higher incidence of CCS class IV (12.7% vs. 22%; p = 0.0001)” is clearly inconsistent with what is presented in Table 1.
  2. Although the authors have the discretion to define abbreviations, I recommend using the conventional abbreviation CPB for cardiopulmonary bypass rather than CBP, as it aligns with standard practice.
  3. Page 5, lines 166–168: The authors state, “Patients who underwent CABG procedures during the COVID-19 pandemic exhibited higher odds of 30-day mortality (OR, 1.9; 95% CI, 0.96–3.67; p = 0.063) and in-hospital mortality (OR, 2.35; 95% CI, 1.15–4.82; p = 0.017).” However, this description is entirely inconsistent with the outcome data presented in Table 3 and with the conclusions. I do not understand where these figures in the manuscript narrative originated.
  4. In the conclusion, the authors state that the COVID-19 pandemic reduced CABG procedures, prioritizing urgent cases, and that patients undergoing CABG during this period had higher 30-day and in-hospital mortality. However, according to Table 1, patients in the pandemic period had a lower proportion of CCS class IV, a better NYHA functional class, a lower proportion of critical preoperative state, and no statistically significant difference in EuroSCORE II compared with the pre-pandemic group. The severity of patients during the COVID-19 pandemic was clearly not higher than that of the pre-pandemic period. Therefore, where does the higher proportion of urgent cases come from? Was it truly that these patients required urgent revascularization, or was it that, during the COVID-19 pandemic, surgical volumes were reduced and available ward and ICU capacity was limited, leading surgeons to be more inclined to operate earlier on inpatients referred by cardiologists? The authors should address this question in the Discussion section. The authors should also provide data on how many CABG patients during the pandemic period were COVID-19 positive at the time of surgery.
  5. Furthermore, the authors should explain why, given that patient severity was not higher during the COVID-19 pandemic, short-term mortality was higher. Was this due to limited care resources during the pandemic, or could earlier surgical intervention have contributed to increased mortality?
  6. Finally, I recommend that the authors carefully review all tables and ensure that all numbers are consistent with the narrative text and free from typographical errors.

Author Response

Reviewer 1:

“The authors reported a single-center retrospective cohort study from Poland, examining the distribution and outcomes of patients undergoing CABG before and during the COVID-19 pandemic.

I have several comments and questions.

  1. Page 4, line 127: The statement “patients during the pandemic had a higher incidence of CCS class IV (12.7% vs. 22%; p = 0.0001)” is clearly inconsistent with what is presented in Table 1.
  2. Although the authors have the discretion to define abbreviations, I recommend using the conventional abbreviation CPB for cardiopulmonary bypass rather than CBP, as it aligns with standard practice.
  3. Page 5, lines 166–168: The authors state, “Patients who underwent CABG procedures during the COVID-19 pandemic exhibited higher odds of 30-day mortality (OR, 1.9; 95% CI, 0.96–3.67; p = 0.063) and in-hospital mortality (OR, 2.35; 95% CI, 1.15–4.82; p = 0.017).” However, this description is entirely inconsistent with the outcome data presented in Table 3 and with the conclusions. I do not understand where these figures in the manuscript narrative originated.
  4. In the conclusion, the authors state that the COVID-19 pandemic reduced CABG procedures, prioritizing urgent cases, and that patients undergoing CABG during this period had higher 30-day and in-hospital mortality. However, according to Table 1, patients in the pandemic period had a lower proportion of CCS class IV, a better NYHA functional class, a lower proportion of critical preoperative state, and no statistically significant difference in EuroSCORE II compared with the pre-pandemic group. The severity of patients during the COVID-19 pandemic was clearly not higher than that of the pre-pandemic period. Therefore, where does the higher proportion of urgent cases come from? Was it truly that these patients required urgent revascularization, or was it that, during the COVID-19 pandemic, surgical volumes were reduced and available ward and ICU capacity was limited, leading surgeons to be more inclined to operate earlier on inpatients referred by cardiologists? The authors should address this question in the Discussion section. The authors should also provide data on how many CABG patients during the pandemic period were COVID-19 positive at the time of surgery.
  5. Furthermore, the authors should explain why, given that patient severity was not higher during the COVID-19 pandemic, short-term mortality was higher. Was this due to limited care resources during the pandemic, or could earlier surgical intervention have contributed to increased mortality?
  6. Finally, I recommend that the authors carefully review all tables and ensure that all numbers are consistent with the narrative text and free from typographical errors.”

We based on medical records, what we show in our resuts. There is no easy explanation of this phenomenon, and all of this multifactorial reasons could be correct. We were unable to identify exact factors responsible for increased higher pandemic mortality despite of unchanged patients risk profile according to EuroSCORE II.

Response to Reviewer 1

We sincerely thank Reviewer 1 for their careful and thorough analysis of our manuscript, as well as for pointing out the inconsistencies that required correction. 

Below, we address each point in detail and describe the changes made in the revised version.

  1. Thank you for pointing out the discrepancy. The numbers in Table 1 are correct (pandemic 12.7% vs pre-pandemic 22%), and the sentence on page 4 incorrectly stated the direction. We have rebuilt the following sentences:

"However, patients during the pandemic had a higher incidence of CCS class IV (12.7% vs. 22%; p = 0.0001) but a lower preoperative critical state (17.6% vs. 24.5%; p = 0.001)." into "However, during the pandemic, patients showed a lower incidence of CCS class IV (12.7% compared to 22%; p = 0.0001) and a reduced rate of preoperative critical states (17.6% versus 24.5%; p = 0.001).”

  1. We acknowledge the inconsistency in abbreviations. We have standardised the abbreviation throughout the manuscript. All instances of "CBP" were replaced with "CPB” (… line in the revised manuscript). Additionally, we updated the list of abbreviations to explain "CPB — Cardiopulmonary bypass.”
  2. Thank you for highlighting the ambiguity. This inconsistency in the text arises from unclear data presentation and stylistic errors made by the authors. To rectify authors:
  3. Revise the sentence in the Abstract from „In-hospital mortality risk rose by 2.35 times, despite unchanged patient risk profiles.” to “In-hospital mortality odds rose by 2.35 times, despite unchanged patient risk profiles”
  4. Rewrite the sentence in the 2.2 Statistical analysis section from „Logistic regression estimated the odds of study outcomes, summarised with odds ratios (ORs) and 95% confidence intervals (95% CIs).” to „The odds ratios (ORS) and 95% confidence intervals (95% CIs) were calculated to determine the association between observation period (pandemic vs. pre-pandemic) and 30-day or in-hospital mortality.”
  5. Change the sentence in the Results 3.1.2 section from „Patients who underwent CABG procedures during the COVID-19 pandemic exhibited higher odds of 30-day mortality (OR, 1.9; 95% CI, 0.96 - 3.67; p = 0.063) and in-hospital mortality (OR, 2.35; 95% CI, 1.15 - 4.82; p = 0.017).” to „The odds of mortality within 30 days after surgery and during a procedure-related hospitalisation during the COVID-19 pandemic were 1.9 (OR, 1.9; 95% CI, 0.96 - 3.67; p = 0.063) and 2.35 (OR, 2.35; 95% CI, 1.15 - 4.82; p = 0.017) times higher compared to patients who underwent surgery before the pandemic.”
  6. Modify the Discussion section sentences from „Furthermore, logistic regression analysis revealed that patients who underwent CABG surgery during the pandemic were at a higher risk of in-hospital death (OR 2.35; 95% CI, 1.15 - 4.82; p = 0.017). This finding indicates that the 2.4-fold increase in the risk of in-hospital mortality among patients undergoing CABG during the pandemic was directly related to the adverse effects of the COVID-19 era.” to “Furthermore, data analysis revealed that patients who underwent coronary artery bypass grafting surgery during the pandemic exhibited higher odds of in-hospital mortality (OR 2.35; 95% CI, 1.15 - 4.82; p = 0.017) compared to those before the pandemic period. This evidence indicates that the observed 2.35-fold increase in the chance of in-hospital death among individuals undergoing CABG during the pandemic is directly associated with the detrimental impacts of the COVID-19 era.”
  7. Rewrite the sentence in Conclusion section from „During the pandemic era, the risk of in-hospital mortality increased 2.35 times, even without worsening the risk profile of patients, as indicated by the EUROSCORE II.” to “The pandemic era increased the odds of in-hospital mortality by 2.35 times, even without worsening patients' risk profile, as indicated by the EUROSCORE II.”
  8. We appreciate the notice of these study points.

First, we aim to clarify the potential mechanisms that resulted in a higher proportion of urgent procedures, despite no change in patients’ clinical severity.

  1. Unfortunately, we lack detailed data on the proportion of patients referred for CABG before and during the pandemic who were in-hospital patients directly referred from cardiology wards. This makes it challenging to draw definitive conclusions about whether the rise in urgent cases was due to earlier inpatient referrals by cardiologists. Additionally, there is no evidence to suggest that the higher share of urgent procedures was caused by earlier cardiology qualification. As highlighted in the Discussion, both Polish and international studies have consistently shown a decrease in admissions for acute coronary syndromes during the pandemic. Notably, a recent Polish multicenter study by Jankowska-Sanetra et al. (reference 27) indicated that during the first COVID-19 wave, the percentage of patients referred from cardiology wards for CABG declined from 4.9% to 3.7%.

An aspect that might influence this but has not been discussed extensively so far is the impact of systemic changes related to the COVID-19 pandemic. Like many countries, Poland's National Health Fund and the Ministry of Health issued recommendations that limited or temporarily halted elective surgeries. In this context, the increased proportion of urgent cases may have been an effort to maintain surgical care continuity despite these systemic restrictions.

  1. We regret that detailed, reliable data on individual patients’ SARS-CoV-2 infection status at the time of surgery are not available in our dataset. We have revised the following sentence in the Limitations section from „The main limitation is the lack of precise data on individual patients' COVID-19 infection statuses.” to “The main limitation is the lack of precise data on individual patients' COVID-19 infection statuses; therefore, the potential impact of perioperative COVID-19 infection on outcomes could not be evaluated.”
  2. We thank the reviewer for their insightful question about the factors behind the increased short-term mortality during the COVID-19 pandemic, especially regarding limited care resources and too-early surgical interventions.
  1. first, the reorganisation of intensive care units and the redeployment of highly specialised staff to COVID-19 wards, which likely affected perioperative management and postoperative care.
  2. Second, the perioperative SARS-CoV-2 infection status of our patients was unknown, and potential sequelae of COVID-19 could have negatively influenced recovery after cardiac surgery. Third, during the pandemic, diagnostic pathways for non-cardiovascular diseases were limited, which may have delayed recognition of comorbidities that could also impact postoperative outcomes.

Taken together, these systemic and contextual influences may explain why observed mortality exceeded risk-adjusted expectations. Significantly, however, our study population did not demonstrate a higher incidence of postoperative complications, and the longer duration of mechanical ventilation observed in the pandemic group was most likely related to heightened concerns about airway management and the risk of droplet transmission of SARS-CoV-2, rather than actual clinical deterioration.

Although numerous publications have demonstrated that very early revascularisation in the setting of acute coronary syndromes is associated with increased mortality, these findings are difficult to translate directly to the interpretation of our results. This is because, as the Reviewer has also noted, patients in the pandemic cohort less frequently presented with features of acute coronary syndromes that might otherwise have prompted cardiac surgeons to postpone surgery to reduce operative risk. It is important to emphasise, as we have in the Discussion, that the pandemic cohort included a higher proportion of patients with three-vessel coronary artery disease (TVCAD) who were referred for surgery on life-saving indications, and at the same time, a greater proportion of patients with diabetes mellitus. In line with the existing literature, ESC guidelines, and the FREEDOM trial, such patients are most appropriately treated with surgical revascularisation.

  1. We performed a careful audit of all tables and figure legends, corrected typographical errors and mismatches between text and tables, and ensured consistent ordering of group percentages and p-values. Specific corrections include:
  2. The word center in Conclusion was changed to “centre”
  3. In the sentence “The percentage of patients classified as CCS IV also decreased almost twofold (22% vs 12.2%, p = 0.0001),” we corrected the percentage values from 12.2% to 7%.

Reviewer 2 Report

Comments and Suggestions for Authors

The manuscript entitled „The impact of the COVID-19 pandemic on coronary artery bypass grafting surgery - a retrospective cohort study” presents an interesting point of view on the influence of the pandemic on cardiac surgery procedures.

 The Authors present the original paper. The title is consistent with the problem presented, and we can suspect the article.  In the abstract, the most important information was included and condensed. The authors explain why they start the topic.

Keywords are mostly correct, but note: keywords are supposed to be different from words in the title/not repeated for better search. (delete-  COVID-19 pandemic, coronary artery bypass grafting)

All article is well written. Everything is clear: study groups, presented results, and DISCUSSION. As well Authors did not forget about the limitations of their work. Based on the analysis of results and the literature review conclusions of the Authors are correct.  References - Most of them are current and brand new.

I evaluate this work as a screening one. It has educational value. Authors take care carefully about  Materials and methods and results. In my opinion, the article is fully written, and only some minor revisions are necessary. I would consider adding to the title information „Single center experience”.

Author Response

Reviewer 2:

„The manuscript entitled „The impact of the COVID-19 pandemic on coronary artery bypass grafting surgery - a retrospective cohort study” presents an interesting point of view on the influence of the pandemic on cardiac surgery procedures.

The Authors present the original paper. The title is consistent with the problem presented, and we can suspect the article.  In the abstract, the most important information was included and condensed. The authors explain why they start the topic.

Keywords are mostly correct, but note: keywords are supposed to be different from words in the title/not repeated for better search. (delete-  COVID-19 pandemic, coronary artery bypass grafting)

All article is well written. Everything is clear: study groups, presented results, and DISCUSSION. As well Authors did not forget about the limitations of their work. Based on the analysis of results and the literature review conclusions of the Authors are correct.  References - Most of them are current and brand new.

I evaluate this work as a screening one. It has educational value. Authors take care carefully about  Materials and methods and results. In my opinion, the article is fully written, and only some minor revisions are necessary. I would consider adding to the title information „Single center experience”.”

Response to Reviewer 2

We sincerely thank Reviewer 2 for the positive and constructive evaluation of our manuscript. We are grateful for the reviewer’s recognition of the clarity in presenting our study groups, results, and discussion, as well as the acknowledgement that our conclusions are consistent with the analysis and literature review.

Below, we address each point in detail:

  1. We agree with the reviewer’s recommendation to avoid repeating words from the title in the keywords for better indexing and search optimisation. We have revised the keywords accordingly and removed “COVID-19 pandemic” and “coronary artery bypass grafting” as suggested. Instead, we added the following keywords optimal for PubMed and Scopus indexing: „myocardial revascularisation”, „cardiothoracic surgery”, „cardiac surgery outcomes” and „pandemic impact on healthcare”. Additionally, the keyword "SARS-CoV-2" was changed to "SARS-CoV-2 infection".
  2. We appreciate the suggestion to emphasise that this is a single-centre experience. We have modified the manuscript title to: “The impact of the COVID-19 pandemic on coronary artery bypass grafting surgery: a single-centre retrospective cohort study” to reflect the study design and scope better.

Reviewer 3 Report

Comments and Suggestions for Authors

Although this paper is a retrospective cohort study based on post-hoc investigations, authors comprehensively have analyzed the impact of the COVID-19 pandemic in Poland, particularly the magnitude of the pandemic three waves, on coronary artery bypass surgery, and I therefore deem it worthy of publication.
The design of the cohort group and the statistical methods employed were appropriate. No major issues were identified in the evaluation of the sample population and data, or in the interpretation of the results based on these. However, as this study was based on data from a single institution, further research involving multiple institutions will be anticipated in the future.

Author Response

“Although this paper is a retrospective cohort study based on post-hoc investigations, authors comprehensively have analyzed the impact of the COVID-19 pandemic in Poland, particularly the magnitude of the pandemic three waves, on coronary artery bypass surgery, and I therefore deem it worthy of publication.
The design of the cohort group and the statistical methods employed were appropriate. No major issues were identified in the evaluation of the sample population and data, or in the interpretation of the results based on these. However, as this study was based on data from a single institution, further research involving multiple institutions will be anticipated in the future.”

Response to Reviewer 3

We sincerely thank Reviewer 3 for the positive and encouraging evaluation of our manuscript. We are pleased that the Reviewer found our analysis of the pandemic’s impact on coronary artery bypass surgery in Poland to be comprehensive and the statistical methods and interpretation appropriate.

We fully acknowledge the Reviewer’s important point regarding the single-institution nature of our study. While our data offer valuable insights into how the three pandemic waves affected surgical volume, urgency, and outcomes at our centre, we agree that future research involving multi-centre or national datasets would enhance the external validity and generalisability of these findings. Therefore:

  1. We have handled the limitations in the Discussion section by adding the following statement:

It is a retrospective, single-centre analysis, which makes it susceptible to bias and residual confounding. “These attributes of design naturally limit the external validity and broad applicability of our findings. Variations in hospital capacity, regional pandemic severity, and local clinical protocols may have influenced surgical practices and outcomes elsewhere. Therefore, future multicentre or national data-based studies are crucial to validate our results and deepen the understanding of the pandemic’s impact on cardiac surgery services.”…

  1. Additionally, we changed the word "primary" to "main” in the sentence: “The primary main limitation is the lack of precise data on individual patients' COVID-19 infection statuses; therefore, the potential impact of perioperative COVID-19 infection on outcomes could not be evaluated.”

Round 2

Reviewer 1 Report

Comments and Suggestions for Authors

The authors have partially addressed the previously raised issues, but some inconsistencies remain uncorrected.

As I noted earlier, on page 5, lines 166–169 they wrote: “The odds of mortality within 30 days after surgery and during a procedure-related hospitalisation during the COVID-19 pandemic were 1.9 (OR, 1.9; 95% CI, 0.96–3.67; p = 0.063) and 2.35 (OR, 2.35; 95% CI, 1.15–4.82; p = 0.017) times higher compared to patients who underwent surgery before the pandemic.” This statement is entirely inconsistent with the data presented in Table 3 (Table 3: in-hospital mortality 1.8 vs 3.3, p = 0.996; 30-day mortality 1.4 vs 3.3, p = 0.374). Yet the authors did not revise this section, nor did they revise Table 3. I previously asked them to carefully verify whether the table values were erroneous, and they replied that they had fully corrected them. In light of the inconsistency between the table and the text, which is incorrect—the table or the prose? Given that the paper’s key conclusion is that in-hospital mortality was higher during the pandemic (OR, 2.35; 95% CI, 1.15–4.82; p = 0.017), from where exactly was this figure derived?

Author Response

Reviewer 1: 

„As I noted earlier, on page 5, lines 166–169 they wrote: “The odds of mortality within 30 days after surgery and during a procedure-related hospitalisation during the COVID-19 pandemic were 1.9 (OR, 1.9; 95% CI, 0.96–3.67; p = 0.063) and 2.35 (OR, 2.35; 95% CI, 1.15–4.82; p = 0.017) times higher compared to patients who underwent surgery before the pandemic.” This statement is entirely inconsistent with the data presented in Table 3 (Table 3: in-hospital mortality 1.8 vs 3.3, p = 0.996; 30-day mortality 1.4 vs 3.3, p = 0.374). Yet the authors did not revise this section, nor did they revise Table 3. I previously asked them to carefully verify whether the table values were erroneous, and they replied that they had fully corrected them. In light of the inconsistency between the table and the text, which is incorrect—the table or the prose? Given that the paper’s key conclusion is that in-hospital mortality was higher during the pandemic (OR, 2.35; 95% CI, 1.15–4.82; p = 0.017), from where exactly was this figure derived?” 

Response to Reviewer 1 

We are very grateful to the Reviewer for carefully identifying this important inconsistency between the prose and Table 3, and for bringing it once again to our attention. After a thorough re-analysis of the data, we recognised that the discrepancy arose not from the raw mortality rates themselves, but from errors in the reported p-values and in the labelling of odds ratios in the text. We have now implemented the following corrections to ensure consistency and accuracy throughout the manuscript: 

  1. Table 3 
  • The percentages of deaths shown in the rows “30-day mortality” and “in-hospital mortality” were correct. 
  • However, the corresponding p-values were erroneous. We corrected them as follows: 
  • 30-day mortality: p = 0.026 
  • In-hospital mortality: p = 0.089 
  1. Odds ratios 
  • We rechecked the corerectness of the odds ratios.The correct values are: 
  • In-hospital mortality: OR = 1.88; 95% CI: 0.96–3.67; p = 0.089 
  • 30-day mortality: OR = 2.35; 95% CI: 1.15–4.82; p = 0.026 
  • These values have now been uniformly corrected in all relevant sections of the manuscript. 
  1. Specific textual corrections 
    a) Results (Section 3.1.2): 
    Corrected sentence: 
    „During the COVID-19 pandemic, the odds of mortality within 30 days after surgery were 2.35 (OR, 2.35; 95% CI, 0.96 - 3.67; p = 0.026), and the odds of in-hospital mortality were 1.88 (OR, 1.88; 95% CI, 1.15 - 4.82; p = 0.089) times higher compared to patients who underwent surgery before the pandemic.” 

b) Discussion: 
Corrected text: 
“However, in-hospital mortality nearly doubled during the pandemic (1.8% vs. 3.3%, p = 0.089). Additionally, data analysis showed that patients who underwent coronary artery bypass grafting during the pandemic had higher odds of 30-day mortality (OR 2.35; 95% CI, 1.15–4.82; p = 0.020) compared to those before the pandemic. This evidence suggests that the 2.35-fold increase in the likelihood of 30-day death among individuals undergoing CABG during the pandemic is directly linked to the adverse effects of the COVID-19 era.” 

c) Conclusions: 
Corrected sentence: 
“The pandemic era increased the odds of short-term mortality without worsening patients’ risk profiles, as indicated by the EUROSCORE II.” 

d) Abstract: 
Corrected sentence: 
“Short-term mortality odds rose, despite unchanged patient risk profiles.” 

All of these corrections have been incorporated into the revised manuscript and highlighted in colour to assist the Reviewer and Editors in their identification. 

We sincerely thank the reviewer once again for pointing out this inconsistency. The manuscript has now been thoroughly revised to ensure the correct values are consistently reflected throughout the text, tables, and abstract. We believe these corrections have addressed the issue and enhanced the clarity and reliability of our findings. 

Round 3

Reviewer 1 Report

Comments and Suggestions for Authors

The authors have addressed all the issues I previously raised, and I have no further comments.